# Physical Activity Component of the Greek Interventional Geriatric Study to Prevent Cognitive Impairment and Disability (GINGER): Protocol Development and Feasibility Study

**DOI:** 10.3390/healthcare12222282

**Published:** 2024-11-15

**Authors:** Evdokia Billis, Eftychia Nastou, Sofia Lampropoulou, Maria Tsekoura, Eleni Dimakopoulou, Nikolaos Mastoras, Ioanna-Maria Fragiadaki, Eleftherios Siopis, Nikolaos Michalopoulos, Paraskevi Sakka, Maria Koula, Maria Basta, Panagiotis Alexopoulos

**Affiliations:** 1Laboratory of Clinical Physiotherapy and Research, Department of Physiotherapy, School of Health Rehabilitation Sciences, University of Patras, 26504 Patras, Greece; eutuxianastou@gmail.com (E.N.); lampropoulou@upatras.gr (S.L.); mariatsekoura@upatras.gr (M.T.); left.siopis@gmail.com (E.S.); nm.michalopoulos@gmail.com (N.M.); 2Patras Dementia Day Care Center, Corporation for Succor and Care of Elderly and Disabled-FRODIZO, 26226 Patras, Greece; panos.alexopoulos@upatras.gr; 3Athens Association of Alzheimer’s Disease and Related Disorders, 15123 Maroussi, Greece; elenidimakopoulou@gmail.com (E.D.); info@psakka.gr (P.S.); 4Ioannina Dementia Day Care Centre, 45221 Ioannina, Greece; nimastoras@yahoo.gr (N.M.); koula_mar@yahoo.com (M.K.); 5Day Care Center for Alzheimer’s Disease PAGNH “Nefeli”, University Hospital of Heraklion, 71500 Crete, Greece; iwanna_fra@hotmail.com (I.-M.F.); mbasta73@gmail.com (M.B.); 6Department of Psychiatry, University Hospital of Heraklion, Medical School, University of Crete, 71003 Heraklion, Greece; 7Mental Health Services, Patras University General Hospital, Department of Medicine, School of Health Sciences, University of Patras, 26504 Patras, Greece; 8Global Brain Health Institute, Medical School, Trinity College Dublin, The University of Dublin, D02 X9W9 Dublin, Ireland; 9Department of Psychiatry and Psychotherapy, Klinikum rechts der Isar, Faculty of Medicine, Technical University of Munich, 81675 Munich, Germany

**Keywords:** physical activity, subjective cognitive decline, feasibility, Greek, SCD, exercise program

## Abstract

Background/Objectives: Individuals with subjective cognitive decline (SCD) have an increased risk of developing dementia, while non-pharmacological multicomponent lifestyle interventions are recommended for prevention/management. The Greek Interventional Geriatric Initiative to Prevent Cognitive Impairment and Disability (GINGER) is such a multicomponent approach, encompassing simultaneous interventions (cognitive training, depression and sleep management, etc.). Exercise/Physical activity (PA) is suggested as one such intervention. This study (i) presents the exercise protocol developed for GINGER and (ii) explores its feasibility (acceptability, applicability, adherence, users’ satisfaction and reliability). Methods: Exercise/PA protocol development, targeting SCD individuals aged > 55 years, utilized relevant guidelines/literature followed by focus group involving exercise specialists. Data were synthesized through consensus to design optimal exercise interventions prescribed on participant’s physical capacity (heart rate, exertion, etc.), comprising 6-month combined aerobic, strengthening, balance and dual-task exercises, delivered 3 times/weekly in two group-based supervised sessions (in-person and online) and one home-based session. Physical outcomes include balance, aerobic capacity [2-Minute Walk Test (2 MWT), IPAQ-7], strength [Hand Grip Strength (HGS), Sit-to-Stand], fear of falling. Eligibility for entering intervention is low IPAQ-7, 2 MWT or HGS scoring. Feasibility was explored with adherence (exercise diaries and Exercise Adherence Rating Scale) and satisfaction (Patient Satisfaction Questionnaire) Results: Intervention was easily delivered with good reliability across testers’ assessments on 13 SCD participants (ICCs = 0.62–0.99), and improved physical outcomes, whereas users’ adherence and satisfaction scored highly. Conclusions: The exercise protocol for SCD was feasible, acceptable, applicable, reliable, demonstrating adherence and satisfaction, while improving physical parameters. It is thus integrated in the GINGER study, where multiple simultaneous interventions will take place to prevent/enhance cognitive function.

## 1. Introduction

Dementia is currently a major public health challenge, expected to test the resilience of health systems internationally in the upcoming years. It is estimated that the number of people with dementia worldwide will be over 131 million in 2050 [1,2]. Since dementia embodies one of the most common causes of disability in older people, it becomes an unprecedented challenge for health systems [3,4,5,6,7]. In Greece, the number of people with dementia is expected to rise from 206,000 in 2019 to approximately 300,000 in 2050, an increase of approximately 45% [1]. The importance of strategies and great efforts to prevent or delay the onset of dementia is thus understood.

The Greek Interventional Geriatric Initiative to Prevent Cognitive Impairment and Disability (GINGER) aims to adopt multilevel lifestyle modification interventions to reduce the risk of dementia in a Greek population. In particular, the focus of the study will be on people who visit memory clinics, psycho-geriatric clinics or other services due to self-experienced complaints regarding their cognitive abilities, such as memory deficits (compared with their previously normal status), while objective examination of their mental and cognitive function through standardized neuropsychological tests is normal for their age. Usually, these people are given the diagnosis of subjective cognitive decline (SCD) and are estimated to account for 25% of the general population aged over 50 years old [8], though prevalence can rise of up to 55% in people over 65 years old [9]. SCD presents a group with tremendous clinical interest, as it is in an in-between state of normal aging and mild cognitive impairment (MCI), while the literature has revealed that it has an increased risk of developing MCI and/or dementia [10,11,12,13]. A 7-year cohort study including SCD and non-SCD (healthy) individuals over 40 years old revealed a 4.5-fold increase in the risk of developing MCI and a 6.5-fold increase in developing dementia, an over 60% more rapid estimated rate of decline compared to those without SCD [14]. Along with the mental deterioration, physical functioning is also reported to be compromised in several areas relating to motor execution, such as gait, physical function, etc., compared to healthy samples [15,16]. Unfortunately, for SCD as well as MCI populations, there are no pharmacological strategies to reduce/prevent this risk of deterioration. Instead, multicomponent interventions targeting several modifiable factors simultaneously (such as depression, physical inactivity, vascular risk factors, sleeping problems, etc.) are considered most appropriate for addressing both mental and physical function [17,18].

One of the recommended interventions within this multidomain ‘hub’ for reducing the risk of dementia in both SCD and MCI populations is suggested to be physical activity (PA) or physical exercise (PE), when delivered in a more structured and target-related mode. PA has recently emerged as a non-pharmacological strategy to reduce age-related cognitive decline and prevent the development of full-blown dementia in SCD and MCI populations [19,20]. Evidence from animal and human studies highlight the positive effects of PE on cognitive function such as memory deficits, attention issues, executive function, through a series of complex underlying mechanisms, including the facilitation of neuroprotective neurotrophic factors, brain plasticity, mitochondrial modulation, cytokine release and other mechanisms [21,22,23]. It is established that physical exercise, specifically incorporating aerobic exercise, is positively associated with cognition [24,25,26,27,28,29]. Strength and balance training can improve executive function and gait parameters [30,31]. Progressive strength and balance training in older people, with or without dementia, reduces risk of falling and reduces carers’ strain, thus improving quality of life, mood and confidence [32,33,34]. Moderate-intensity exercise, 2–3 times a week, improves strength, gait speed, and performance on activities of daily living [35,36]. There may be additional benefits in slowing cognitive decline, although the size of this effect still appears to be small [29,32].

Indeed, various forms of aerobic and strengthening exercise have been applied to people with MCI, with positive effects on their physical and mental well-being [27,28]. Within SCD populations, although there is still very little research into optimal exercise protocols, evidence is encouraging in that PE improves neurophysiological activities associated with memory function and executive function, let alone the physical and functional benefits [28,37]. Furthermore, encompassing PE programs in multicomponent interventions is suggestive of overall mental and physical enhancement and is thus highly recommended [17,18].

Given the above, the objective of this study is to develop a study protocol for a PE progressive program that could be integrated in a multicomponent intervention pathway (GINGER) targeting people with SCD. Thus, this paper aims firstly to present the developed protocol for the exercise program (PA component) of GINGER for people with SCD, and secondly to explore its feasibility in terms of acceptability, applicability, adherence and satisfaction to SCD users as well as the reliability of its implementation.

## 2. Materials and Methods

### 2.1. Protocol Development

Τhe GINGER intervention for people with SCD has been designed by healthcare professionals of the following institutions: The Old-Age Psychiatry Outpatient Clinic of the General University Hospital of Patras, the Department of Physiotherapy of the University of Patras, the Alzheimer Athens Day Care Centers, the Alzheimer’s Day-Care Center Nefeli at the General University Hospital of Heraklion, the Department of Neurology of the University of Crete and the Day Care Center in Ioannina run by the Society of Psychosocial Research and Intervention (EPSEP). The GINGER intervention encompasses cognitive training, sleep management, dietary modifications, hearing/vision support, intervention for depression, management of metabolic and vascular risk factors, and physical activity interventions, and a combination of 2–4 interventions (according to each subject’s indications) are organized to take place simultaneously (multicomponent nature of the intervention). Ethical approval has been obtained from the Ethical Committees of the University of Patras and the University Hospital of Patras and the current trial has been registered at ClinicalTrials.gov (NCT06528379). The PA component of GINGER is coordinated by staff of the Department of Physiotherapy of the University of Patras.

Intervention participants are recruited from the aforementioned centers, whereas the PA component is coordinated by the Patras site. Inclusion criteria are people aged over 50 years old who complain about memory deficits confirmed by the SCD Questionnaire score (SCD-Q > 7) [38] and have normal mental status after examination performed by memory professionals, utilizing Montreal Cognitive Assessment (MoCA > 26) [39]. Exclusion criteria include (i) people suffering from chronic mental or neurological disorders or unstable pathological diseases affecting mental function (e.g., schizophrenia, bipolar disorder, multiple sclerosis, history of traumatic brain injury, hydrocephalus, Parkinson’s disease, epilepsy, thyroid disorders), (ii) people with insufficient knowledge of the Greek language, (iii) people with severe sensory disorders that make communication impossible, and (iv) people who have had recent surgery.

#### 2.1.1. Development of the Physical Activity Assessment and Intervention Procedure

A comprehensive literature review was conducted, followed by a focus group study. The literature review focused on assessing physical activity procedures and tools, guidelines, and exercise interventions for individuals with SCD, MCI, dementia, as well as healthy community-dwelling elderly individuals. Given the age of the target population (over 50), cardiovascular parameters were also considered. As a result, the search for exercise intervention programs was directed towards cardiovascular rehabilitation and the exercise protocols applicable to these demographic populations. Various combinations of keywords were utilized to identify relevant exercise protocols, including terms such as physical exercise, protocols, rehabilitation, aerobic, strength training, balance, dual-tasking, cardiovascular disease, assessment tools, functionality, falls, and adherence. Search results indicated that a structured aerobic exercise program of moderate to high intensity, including activities like walking, cycling, or dance routines, is beneficial for all participants, enhancing cardiovascular health, aerobic capacity, balance, mobility, muscle strength and cognition, with the most significant improvements in cognitive speed, memory, and attention [35,40,41]. Resistance exercises also boost muscle strength, power, and muscle mass while helping to reduce falls [42]. Combining resistance and aerobic exercises further improves cardiovascular health and aerobic capacity [40]. Various forms of exercise have been shown to improve physical functioning in older adults with dementia, regardless of the disease’s progression, with the best outcomes linked to programs of higher training volumes. [43]. In addition, the World Health Organization advises engaging in 150 min of moderate-intensity or 75 min of vigorous-intensity physical activity each week. Furthermore, regular exercise, performed 2–3 times a week, enhances the physical condition of older adults and regulates their vascular health [42]. Also, supervised exercise led by professionals is found to be generally more effective than unsupervised workouts and exercising in small groups tends to yield better results than individual (one-to-one) programs [44].

Following this literature summary, a focus group was organized, involving four health professionals and one moderator/coordinator, all physiotherapists (PTs) with over ten years of clinical experience in exercise therapy. One of the PTs was also highly experienced in treating people with MCI and dementia, whereas the other three had extensive clinical experience with exercise therapy (>10 years) across elderly populations. The moderator also had prior experience with focus groups [45,46]. Following a small presentation to the group by the moderator clarifying research purposes, the issues discussed within the focus group were as follows: (1) to select a list of valid and suitable outcome measures for the targeted population, (2) to agree on which tests will be used for determining which participants will enter the exercise intervention, and (3) to develop the key exercise sets included in the intervention program. Main lists (tests, outcomes, exercises etc.) were written down in a flowchart by the moderator as the discussion was developing and in the second part of the focus group, these lists were reconsidered to reach consensus. The focus group ran smoothly, lasting for 3.5 hrs; intra-group agreement was achieved, and all aims were satisfied.

#### 2.1.2. Outcome Measures

Following the focus group, six outcomes representing balance, muscle strength, physical function, fear of falling and aerobic capacity were selected for the PA component. Although cognitive function forms the primary outcome measure for the GINGER study, this was not tested here, given that only one intervention—physical exercise—was tested in the current protocol and feasibility study. Cognitive performance is tested when more than one intervention targeting different domains is simultaneously being delivered within the GINGER multicomponent study (as indicated in Section 2.1). All physical outcomes selected were representative for older people and had established cutoffs and normative data across age ranges and sex. The selected outcomes were as follows.

Mini-BESTest. The Greek version of the Mini-BESTest, a reliable and valid balance assessment tool [47,48,49], focuses primarily on dynamic balance, containing 14 different tasks, each scoring from 0 to 2, while overall scores range from 0 (lowest function) to 28 (highest level of function) and subsequent subscales score as Anticipatory (0–6), Reactive Postural Control (0–6), Sensory Orientation (0–6) and Dynamic Gait (0–10).

Hand Grip Strength (HGS). HGS is an objective measure of upper limb muscle strength, applied via electronic hand dynamometer (Jamar) [50]. The procedure (squeezing dynamometer as hard as possible) is performed three times on each participant’s hand flexed at 90° at the elbow, while seated in a standard 45 cm height chair without an armrest, under a standardized protocol [51,52].

Sit-to-Stand test. Lower limb strength will be measured using the 30 s Sit-to-Stand test [53]. Participants are seated in a standard chair with their hands crossed in front of their chest. They are asked to stand up completely (full extension of their knees) and sit back down in the chair as many times as possible in 30 s, counting the number of repetitions.

2 Minute Walk Test (2 MWT). The 2 MWT assesses the participants’ aerobic capacity [54]. Each participant is placed in a 9 m hallway and is asked to cover as much ground as possible in two minutes, counting the distance covered.

Functional Efficacy Scale International (FES-I). The Greek version of the FES-I is used to evaluate fear of falling [55]. This popular 16-item self-administered questionnaire scores from 16 to 64 points, where scores between 16–19, 20–27 and 28–64, indicate low, moderate and high concerns about falls, respectively.

International Physical Activity Questionnaire (IPAQ-7). The Greek IPAQ-7 is the short version of the IPAQ questionnaire, evaluating physical activity [56]. It is a 7-item self-reported tool calculating the physical activities taking place during the last week, which are categorized into three levels: low-, moderate- or vigorous-intensity activities, based on duration, days of exercising and metabolic equivalent of tasks (METs). The IPAQ-7 items are structured to provide separate scores on walking, moderate-intensity and vigorous-intensity activities. Computation of the total score requires summation of duration (minutes) and frequency (days) of walking, moderate-intensity and vigorous-intensity activities, being expressed as MET-min per week (MET level × minutes of activity/day × days per week). As a cut-off threshold, the MET score of 600 was used, as below this score, low activity is indicated [57].

#### 2.1.3. Screening Process

For eligibility in the PA component of the trial, a screening process takes place, where participants are assessed for their physical function based on a quick set of representative tests; should participants score less than expected on any of the tests (based on the established normative/cut-off values), then they are eligible to enter the PA intervention. From the focus group, it was decided to include the following three (of the six outcomes) for screening: the HGS, the 2 MWT and the IPAQ-7.

#### 2.1.4. Physical Exercise Intervention

Following the focus group consensus, the PA intervention is outlined below.

Delivery mode. The 6-month intervention includes three weekly exercise sessions, two group-based supervised ones (in groups of up to 5 participants) and a home-based (individualized, non-supervised) one. One group session is delivered live in the exercise laboratory setting of the Department of Physiotherapy (Patras site) and the other one via teleconference (in real time) for facilitation of attendance.

Exercise program and intensity. The exercise program comprises a combination of aerobic exercise, resistance training, balance and dual-task training, according to official guidelines and evidence from similar population samples (SCD, MCI, dementia and elderly). A typical live or online exercise session is structured, encompassing warm-up (5 min), cool-down (5 min), and 40–50 min exercises. The warm-up part comprises walking on the spot for a couple of minutes and then combining it with arm movements, whereas cool-down comprises walking on the spot for a minute and stretching of big muscles (i.e., quadriceps, hamstrings, triceps brachialis, pectoralis major, etc.). The main exercise set comprises aerobic exercise (AE), resistance exercise (RE), balance exercise (BE) and dual-task activities. AE is of moderate to high intensity, 60–85% of targeted heart rate (THR) calculated based on Karvonen’s formula [58] utilizing a smart watch, given to each participant for monitoring 60–85% of their THR. For the RE and BE exercises (some of which are combined), there is a cycle intervention program (with pauses, repetitions and sets), targeting large muscle groups. Repetitions of each RE are performed at a moderate intensity of Borg Rating of Perceived Exertion (RPE) [59]. Thus, the intensity of each RE should be characterized as “somewhat hard” (4) to “very hard” (7) on the 10-point scale of Borg Rating of Perceived Exertion (RPE) [59]. Thus, exercise intensity is individually tailored. For each BE, up to 10 repetitions are performed with 5–10 s holds (gradually progressing). For the home-based program, careful instructions are given to individually perform a chosen aerobic activity (i.e., walking, cycling) for approximately 30 min/week prescribed at a medium intensity (60–80% THR). Distance, heart rate, walking pace, etc., are recorded with a smart watch and fed back to a PA e-platform which is monitored by the therapists. The program and key set of exercises are summarized in Table 1. The exercises are the same for every participant but the progression of each exercise depends on the subject’s physical performance as well as RPE.

Monitoring progress and adherence. Progress tracking is monitored every 3–4 weeks, based on Borg RPE; scoring less than 6/10 determines exercise progression. Exercise adherence is assessed on a predefined weekly exercise diary (exercise log) and the 3-month completion of the Exercise Adherence Rating Scale (EARS) [60,61]. EARS is composed of six items assessed via a 5-point Likert scale, whose possible sum scores range from 0 to 24, where higher sum scores indicate greater exercise adherence. Efforts to improve compliance include a weekly reminder via SMS or phone, communication with the therapist in cases of two consecutive absences and regular feedback on individual progress and achievements in relation to his/her physical condition.

Re-assessment and follow-up. Evaluation takes place at 3 and 6 months (end of intervention), utilizing the aforementioned outcomes in random order. A 6-month follow-up is also scheduled to take place.

#### 2.1.5. Instructors

Experienced physical therapists and exercise scientists, following appropriate training on the assessment/intervention procedures under the supervision of the coordinating site (Patras), deliver the whole program.

### 2.2. Feasibility Study

A feasibility study took place in the coordinating site (Patras), aiming to (1) assess the inter-tester reliability of the assessment procedures across the exercise testers, (2) familiarize/train the instructors on the safe application of the intervention procedures and (3) evaluate the applicability, adherence and satisfaction of the intervention to the users.

For reliability testing, the principal investigator presented the assessment procedures to the testers following 3 h practical training, including demonstration and practice by four testers, two physiotherapists and two physical education teachers (all with multi-year experience in exercise prescription/delivery of MCI and dementia populations). After sufficient training, testers were randomly divided into two groups of two, to perform the assessments. Consecutive older adults who met the eligibility criteria for PA intervention at the Patras site (between April and May), participated in the reliability procedure. Mini-BESTest, 2 MWT, and Sit-to-Stand tests were the outcomes tested for reliability. They were performed once by each participant following standardized instructions by one of the testers, while independently measured participant performance. The sequence of each test was random for each participant and a 5–10 min resting period between tests was given.

Delivery of the treatment protocol was presented by the principal investigator with video-taped exercises, PowerPoint and live model demonstrations. Extensive discussions and solutions to foreseeing problems took place (i.e., regarding exercise progressions, testing site variability, teleconferencing sessions, etc.) and a common drive with important information (exercise tips, handouts, forms, printed programs, etc.) was uploaded for assistance at this stage. Finally, exercises (set at particular intensities based on Kanvonen’s formula and RPE) were practiced/tested in pairs and the principal investigator advised/corrected accordingly. For both the online and real-setting interventions, a series of 8–10 guided training sessions (each) were undertaken by the instructors to familiarize themselves with the procedures, get accustomed to camera settings, time management with exercise repetitions, finalize verbal instructions, safely deliver the exercises, etc.

Participants from the reliability study who fulfilled at least one of the three screening criteria were invited to participate in the intervention protocol, which lasted for 3 months and was undertaken by two of the trained PTs (Patras site). Participation was preceded by a cardiological check-up, which ensured each participant’s safe involvement in physical exercise activities. Adherence was monitored through an exercise diary, and at 3 months, EARS was administered. The Client Satisfaction Scale (CSQ-8) in Greek was administered to evaluate satisfaction on the service provided; CSQ-8 scores range from 8 to 32, where higher scores indicate higher levels of satisfaction [62]. Additionally, participants were encouraged to discuss any problems or concerns at any time throughout the intervention.

Descriptive analysis (means, standard deviations, etc.) were reported. Reliability results were calculated with intraclass correlation coefficient (ICC2.1) two-way random effects model, standard error of measurements (SEM) and smallest detectable differences (SDD), utilizing SPSS statistical package (version 28.0). Paired sample *t*-tests calculated any differences at 3 months across the outcomes measured.

## 3. Results

Results concern the feasibility study, as the protocol study has been detailed in Methods.

### 3.1. Reliability Testing

Overall, thirteen older adults (four men, nine women), aged 65.92 ± 8.6 years (range 56–78), with MoCA score 26.54 ± 1.56, participated in reliability testing, all of which were referred from the Old-Age Psychiatry Outpatient Clinic (General University Hospital of Patras). Descriptive analysis for any one tester pair is summarized in Table 2 and inter-tester reliability of a given pair is provided in Table 3. Reliability across pairs of testers on the outcomes assessed yielded very good to excellent results, with ICCs ranging between 0.62 and 1.00 and SEM and SDD ranging between 0.19–8.71 and 0.52–29.90, respectively.

### 3.2. Familiarization/Testing of the Intervention

Training of the health professionals/physical educators ran smoothly and proved necessary, as sufficient familiarization and practical training with the exercise sets were considered important, as well as adequate understanding of exercise prescription and progression procedures. In addition, familiarization of the teleconferencing mode was important for achieving optimal camera positioning and voice control as well as practice feedback and correction strategies through virtual backgrounds.

### 3.3. Protocol Delivery and Applicability

Seven out of the thirteen SCD participants referred and included in reliability (Section 3.1) did not require PA intervention (based on the three screening tests). Out of the six eligible patients, three, one man and two women, aged 70 ± 12.2 (range: 56–78 years) with MoCA 25.67 ± 1.53, completed the intervention. The remaining three did not enroll due to personal reasons. In particular, one participant had scheduled an eye operative procedure, the second one had a family bereavement and refused participation, and the third one had scheduled a 2-month transatlantic trip, and thus, all were unable to enroll in the delivery of the study.

The sample’s profile, baseline and 3-month outcome measures are summarized in Table 4. No adverse events associated with the exercise intervention were reported. Some minor problems during the first month related to the familiarization of one of the users with the teleconferencing mode were solved. No other problems emerged throughout the intervention.

### 3.4. Timeline Evaluation and Data Loss

All assessments were performed in one contact (approx. 40 min) following the 3-month intervention. There were no missing data on outcomes.

### 3.5. Exercise Adherence and Satisfaction

Records from the participants’ exercise diaries reported 100% engagement with the 3-month intervention by two and 83% engagement by one, as she failed to attend two group sessions. All three participants fully adhered to the home-based program. EARS was highly scored (mean: 27.67 ± 4.51 range: 23–32). Participant satisfaction with the service offered was highly scored on the CSQ-8 (mean: 29 ± 1.7, range: 28–31), indicating significant satisfaction.

## 4. Discussion

This report presents the study protocol of an intensity-monitored and well-developed exercise program for SCD participants, which appeared feasible, safe, reliable, easy to administer, showed compliance and satisfaction to a small user sample, while providing improvements in several physical performance variables. This exercise program is suggested for integration in the GINGER study, a multicomponent lifestyle modification intervention (PE component is one of these) aiming to reduce the risk of dementia amongst older people with SCD living in Greece. As SCD populations are known to have an increased risk of developing dementia [10,11,12], and, as there are no pharmacological strategies to reduce this risk, multimodal, non-pharmacological interventions are considered of primary importance, when run simultaneously, for improving cognitive function as well as for maintaining or improving their physical and mental well-being [29,37,63,64].

Interestingly, although PA and exercise are highly recommended in people with dementia or MCI, few exercise intervention studies have engaged SCD populations [29]. Early management strategies are strongly emphasized for being of utmost importance in preventing brain aging and cognitive degeneration [19,65,66,67]. Thus, since cognitive decline, mental and physical well-being benefit from exercise interventions in people with MCI or dementia, it is not unreasonable to assume that SCD populations might benefit, too.

The mechanisms of the beneficial impact of PA on preventing dementia are complex and not thoroughly understood yet, though PA and exercise may have direct and indirect effects on cognitive function and overall brain health [29,36,68]. A recent body of evidence from both animal and human studies supports molecular-based mechanisms of action being enhanced with exercise, including the improvement of neurotrophic factors in the brain (neurogenesis–angiogenesis–synaptogenesis), increase in hippocampal volume, reduction in biochemical markers of inflammation in the brain, higher neuronal efficiency, mitochondrial biogenesis, etc., all enhancing global cognitive function, attention, processing speed, memory and verbal fluency [23]. Furthermore, exercise reduces vascular risk factors, modulates glucose metabolism and resistance to insulin, which indirectly delays cognitive impairment [20,23,65]. Additionally, psychologically induced mechanisms of exercise, such as improvements in mood, anxiety, depression, sleep quality, stamina, etc., are also found to indirectly impact coexisting neuropsychiatric symptoms, and appear important in brain health [22].

Although aerobic exercise alone has been proposed as the exercise mode for improving cognitive executive function and physical functionality [29], combinations of aerobic, strengthening, balance and dual-tasking exercises over recent years are believed to have additional benefits in terms of fall prevention, muscle strengthening, physical fitness, sarcopenia limitation, cardiovascular benefits, promoting independence, etc. [62,68,69,70]. Indeed, the exercises included in this protocol utilize such a combination with a greater proportion of AE per session. Exercises were carefully chosen from the literature and the focus group consensus (as indicated). According to Kraemer et al. [71], 150 min of moderate-to-vigorous aerobic weekly exercise is recommended for adults over 65 years. The resistance training program is based on evidenced literature [72,73], whereas progression principles are outlined in guidelines of the American College of Sports Medicine [74]. For balance training, the Otago program was found effective in older adults [75,76,77] and dual-task training is promising for physical and cognitive issues [42]. Exercise duration was set three times weekly for 6 months, which is an acceptable and feasible timeline for physical and cognitive changes to take place [35,36]. Exercise intensity was set at moderate levels, based on Karvonen’s formula and Borg’s RPE, as recommended [40,60,78]. This criterion was also important for satisfying a more individually based approach, as exercise precision, specificity training and custom-based prescription are important principles when designing exercise programs [79]. Group-based exercise was also chosen because of its superiority (compared to one-to-one sessions) in physical performance and psychosocial benefits amongst the elderly [35,44,75]. In order to maintain good supervision, small groups were considered ideal. Supervised sessions were preferred, as they are in general more effective than unsupervised ones [44]. Additionally, one individualized home-based session was considered necessary for motivating and maintaining mobility of the participant, while at the same time facilitating exercising in his/her home or another preferred environment [80]. Regarding the group exercise delivery, one of the two weekly supervised sessions is delivered online. In view of the equally good engagement and effectiveness of teleconferencing exercise sessions compared to in-person ones, as well as the ease in delivering an exercise program in real time from the comfort of the person’s home (thus avoiding transportation), this was considered practical, useful and necessary in order to maintain adherence to the exercise protocol [81]. Thus, this designed exercise program is believed to be well thought out and reasoned, aiming to address physical and wellbeing issues amongst SCD populations; thus, we think it is worth integrating into the GINGER multicomponent study.

Outcomes selected for the exercise protocol were all representative for evaluation of physical parameters of elderly mental health populations. Additionally, the three screening tools selected for entering the PA intervention (HGS, 2 MWT IPAQ-7), apart from having well-established cutoffs across all age ranges (as indicated in Section 2), were quick and easy to apply (5–7 min in total).

The feasibility study aimed to explore the overall acceptability, applicability, adherence and satisfaction of the intervention to the users as well as the reliability of assessment. Results provide evidence that the assessment/screening procedures were reliable and that, based on the adherence and satisfaction scales’ results, the intervention was well delivered by the instructors and was acceptable and applicable for the participants. The small number of eligible participants was a shortcoming; however, given the limited timeline and the fact that not all SCD populations present with PA deficits, it was considered acceptable to proceed. Nevertheless, in terms of the GINGER protocol, it should be appreciated that, to recruit 200 for the exercise intervention, a much larger number of participants will be needed.

Exercise adherence was high; it was excellent in two and very good in one participant, based on exercise diary documentation and EARS. Absence was kept to a minimum, which could be attributed to the user-friendly scheduled exercise intervention. Only one out of the three weekly sessions were scheduled in a clinic-based environment; the other one was the home-based online program and the third was the PA task (walking) undertaken in each participant’s optimal choice of environment. Thus, this organization setting was considered facilitatory for attendance. Additionally, the combination of two supervised sessions with a home-based one seemed appropriate. There were two supervised sessions, compared to only one unsupervised one, as they appear more effective than the later (unsupervised) ones [44] In addition, supervision helped participants to comprehend and be reminded weekly of the correct execution of the prescribed exercises, be corrected when needed and safely execute them, while the unsupervised home-based session assisted in self-motivation and appreciation of a more self-paced approach. In addition, the home-based and the online session assisted in adherence to the exercise, as participants did not have to plan for transportation. Also, virtual exercise mode did not cause them any problems in terms of internet connection, exercise application or quality of exercise delivery. However, familiarization time with the internet/online modalities was necessary, and it is recommended ensure the availability of these modalities prior to commencement of an exercise program. The repeated scheduled reminders and communications as well as the regular feedback from the therapist regarding participant progress were considered useful adjuncts for complying with the exercise. Such reminders appear to enhance adherence to exercise and are recommended.

Based on the post-intervention re-evaluations, and despite the shorter exercise protocol delivery, all assessed outcomes improved at 3 months. Interestingly, three variables (FES-I, IPAQ-7 and Sit-to-Stand test) reached significance, while two others (HGS-right hand and anticipatory reaction in MiniBEST) were close to being statistically significant despite the limited number of participants. Cognitive function was not evaluated for several reasons; firstly, the multicomponent intervention nature of GINGER is what is believed to enhance cognitive function, and this protocol had only one component. Secondly, the scope of this study was to develop the exercise protocol and test its feasibility in terms of applicability, user adherence and satisfaction. Thirdly, the 3-month (unicomponent) intervention was considered too short for such changes to take place.

However, considerable strengths of this study lie in the design as well as the delivery mode and intensity parameters of the exercise intervention; the multimodal, progressive, flexible and hybrid-delivered nature of this exercise program, in combination with the fact that it is tailored to the physical capabilities of each individual (i.e., heart rate, RPE, etc.), is strongly believed to provide physical, psychosocial and cognitive benefits as well physical and mental wellbeing.

Limitations of the feasibility study include the shorter duration of the intervention (3 instead of 6 months) due to serious time constraints as well as the lack of reported cognitive outcomes (as previously described). Unfortunately, this tight timeline not only prevented further recruitment of more participants but also long-term follow-up on physical performance. However, the primary aim of this feasibility study was the familiarization and safe delivery of the assessment/intervention procedures by the therapist and participants’ adherence and satisfaction. As adherence was satisfactory amongst participants, it has been assumed that compliance with this flexible exercise program being offered will continue. Nevertheless, as this study is a very small-scale feasibility study, it is necessary to deliver the exercise program across larger SCD participant samples, along the whole 6 months and in parallel with other interventions (i.e., cognitive training, depression management, etc.), to be able to explore and generalize on physical wellbeing, mental wellbeing and cognitive function.

## 5. Conclusions

This study developed a tailored exercise intervention for SCD participants. It has shown to be feasible, safe, reliable, easy to administer, good compliance and satisfaction from its users and yielded improvements in several physical performance variables (balance, strength, aerobic capacity, fear of falling), while intensity parameters, supervision, adherence-motivating acts were monitored. Thus, it is designed for delivery to SCD participants within the Greek healthcare setting as well as integration in the GINGER study, where multiple simultaneous interventions will take place to prevent cognitive decline.

## Figures and Tables

**Table 1 healthcare-12-02282-t001:** Exercise program outline.

Timeline	0–1 Month	2–3 Month	3–4 Months	4–5 Months	5–6 Months
**Aerobic exercise**					
**Exercise mode**					
Frequency/week	3	3	3	3	3
Duration (min)	20–30	20–30	20–30	20–30	20–30
Intensity (RPE)	6	7	8	7	8
**Types of exercises**					
Upper limb exercises	Arm movements (all directions)	Fast-paced arm movements (all directions)	Fast-paced arm movements (all directions) combined with stepping		Fast-paced arm movements (all directions) combined with sideline stepping and knee flexion
Lower limb exercises	Side stepping	Side stepping with boxing	Side stepping and trunk rotations combined with boxing		60 s side stepping and trunk rotations combined with boxing
Dual Task exercises	Side stepping (wide to narrow support base) combined with backward counting		Fast-paced side stepping (wide to narrow support base) combined with backward counting		Fast-paced side stepping (wide to narrow support base) combined with arm movement and backward counting
Core stability/Trunk exercises	High knees	High knees with arm movements	High knees with arm movements and trunk rotations		High knees with arm movements and trunk rotations (i.e., right elbow touches left knee)
**Resistance and balance training exercises**					
**Exercise mode**					
Frequency/week	2	2	2	2	2
Duration (min)	30–45	30–60	30–60	30–60	30–60
Load (RPE)	6	7	8	7	8
Muscle group (number)	8–10	8–10	8–10	8–10	8–10
Rest between sets (min)	1	1	1	1	1
Repetitions/set	8–12	10–12	10–12	10–12	10–12
Number of sets	2	2	2	2	2
**Types of exercises**					
Open Kinetic Chain	Rowing exercise/Hip abduction holding a chair	Rowing exercise holding weight/Hip abduction without holding	Hip abduction without holding	Rowing exercise using elastic band/hip flexion and abduction touching a chair	Rowing exercise using elastic band in pairs/Hip flexion and abduction without holding
Closed Kinetic Chain	Mini squat	Squat	Sumo squat	Squat and calf raise	Sumo squat and calf raise
Dual-Task exercise		Squat combined with shoulder flexion	Squat combined with shoulder flexion and then calf raises		Squat combined with shoulder horizontal abduction using elastic band and then calf raises

**Table 2 healthcare-12-02282-t002:** Sample’s profile (n = 13) and testers’ scores.

	Tester 1		Tester 2	
	Min–Max	Mean (SD)	Min–Max	Mean (SD)
IPAQ-7 score	280–3972	1187.11 (962.40)	-	-
FES-I	16–32	22.85 (4.910)	-	-
HGS—Right (mean of 3 trials)	19.20–48.33	31.49 (9.10)	-	-
HGS—Left (mean)	5.10–43.27	26.61 (11.00)	-	-
**Sit-to-Stand Test (reps)**	10–20	14 (2.90)	10–20	13.92 (3.00)
** 2 Minute Walk Test (m)**	121.77–212.32	167.02 (23.10)	137.63–212.32	170.44 (19.00)
** Mini-BESTest (total score)**	17–27	23.38 (3.40)	17–28	23.38 (3.60)
Anticipatory	3–6	4.62 (0.90)	3–6	4.85 (0.80)
Reactive postural control	3–6	4.85 (1.10)	3–6	4.62 (1.30)
Sensory orientation	4–6	5.69 (0.60)	5–6	5.77 (0.40)
Dynamic gait	4–10	8.23 (1.80)	4–10	8.15 (1.80)

IPAQ = International Physical Activity Questionnaire, FES-I = Functional Efficacy Scale International, Min = minimum, Max = maximum, SD = standard deviation, reps = repetitions, m = meters.

**Table 3 healthcare-12-02282-t003:** Inter-tester reliability results.

	ICC	95% CI (Lower–Upper)	SEM	SDD (%)
** Sit-to-Stand Test (frequency)**	1.0	0.98–1.00	0.19	0.52
** 2 Minute Walk Test (m)**	0.83	0.54–0.95	8.71	14.30
** Mini-BESTest (total score)**	0.96	0.87–0.99	0.70	8.30
Anticipatory	0.62	0.14–0.87	0.51	29.90
Reactive postural control	0.93	0.79–0.98	0.31	18.20
Sensory orientation	0.87	0.63–0.96	0.19	9.20
Dynamic gait	0.96	0.88–0.99	0.35	11.80

ICC = intraclass correlation coefficient, CI = confidence interval, SEM = standard error of measurement, SDD = smallest detectable difference.

**Table 4 healthcare-12-02282-t004:** Baseline and 3-month post-intervention outcomes of the pilot sample.

	Baseline		3-Months		*p*-Value
Outcomes	Min–Max	Mean (SD)	Min–Max	Mean (SD)	
IPAQ-7 score	1257–5931	3009.33 (2546.94)	3139–9492	5434.33 (3524.13)	0.03 *
FES-I	24–30	27 (3)	23–28	25.33 (2.50)	0.02 *
HGS—Right (mean)	22.40–36.60	30.54 (7.30)	22.4–33.03	28.17 (5.40)	0.09
HGS—Left (mean)	19.07–34.87	28.06 (8.10)	19.23–33.63	26.91 (7.20)	0.13
Sit-to-Stand Test (reps)	11–12	11.33 (0.58)	13–15	14 (1)	0.03 *
2 Minute Walk Test (m)	121.80–164.80	137.36 (23.80)	125.10–152.50	142.15 (14.91)	0.38
Mini-BESTest (total)	19–23	21.33 (2.080)	21–25	23 (2)	0.10
Anticipatory	4–6	5 (1)	5–6	5.67 (0.58)	0.09
Reactive postural control	3–4	3.67 (0.58)	3–4	3.33 (0.58)	0.21
Sensory orientation	4–6	5.33 (1.16)	4–6	5.33 (1.16)	0.53
Dynamic gait	7–8	7.33 (0.58)	8–9	8.67 (0.58)	0.09

IPAQ-7 = International Physical Activity Questionnaire, FES-I = Functional Efficacy Scale International, Min = minimum, Max = maximum, SD = standard deviation, Reps = repetitions, m = meters. * statistical significance (*p* < 0.05).

## Data Availability

The datasets presented in this article are not readily available because the data are part of a feasibility and shortly an ongoing study. Requests to access the datasets should be directed to the corresponding author.

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
