# Peer review of "Physical Activity Component of the Greek Interventional Geriatric Study to Prevent Cognitive Impairment and Disability (GINGER): Protocol Development and Feasibility Study"

_healthcare, 2024, doi:10.3390/healthcare12222282_

Round 1

Reviewer 1 Report

Comments and Suggestions for Authors

Dear Authors,

The paper ‘Physical Activity component of the Greek Interventional Geriatric Study to Prevent Cognitive Impairment and Disability (GINGER): protocol development and feasibility study’, presents an interesting question about one physical exercise protocol. However, the objects and especially the justification of the proposal are not clear in the study. This caused difficulty in understanding what the authors want with the study and what applicability of the results found.

GENERAL COMMENTS:

Introduction

Lines 65 – 71

The authors present the definition for subjective cognitive decline (SCD) (people with only self-reported complaints, but without any cognitive decline or dementia tracked in cognitive testing) and present the risk of this population developing dementia. It would be interesting to add how this process develops (how many years on average) to reach dementia and the relationship with mild cognitive impairment (MCI).

Lines 73 – 74

Suggestion: make it clear to the population that the statement about the effect of exercise and the risk of dementia is referring to.

Lines 93 – 96

The authors presented an introduction on the effects of exercise on cognition and the risk of dementia and defined the objective of proposing a physical training protocol. What is the justification for this proposal? This objective should be better contextualized throughout the introduction, making clear the question that the study aims to answer.

Would the objective of the study be to reduce the risk of dementia? Or to modify the lifestyle that can cause, among other things, the risk of dementia?

Material and methods

Lines 113 - 116

The selection of volunteers is unclear. Were people with memory complaints but with normal cognitive performance (assessed by MOCA) recruited? Was there any additional assessment or was the MOCA instrument only applied?

General comment

The authors describe the process of selecting and creating the intervention protocol. However, it is not clear whether there was any basic proposal that had already been tested previously, or what the justification was for proposing a ‘new intervention protocol’. An important issue is that throughout the introduction, only the effects of exercise on cognition are highlighted, but the paper only describes the effects of exercise on physical performance tests. Thus, a latent question of great importance is whether the proposed protocol can modify people’s cognitive performance. Furthermore, it was applied to ‘normal’ people: would people already with cognitive decline be capable of performing the proposed protocol?

Lines 122 - 126

The authors describe that they carried out a literature review to select the tests and develop the physical activity assessment protocol. However, the criteria used to select each test are not presented.

Lines 185 - 209

The authors describe the composition of the protocol and the load variables (intensity, frequency, duration, complexity, etc.). However, after the 3rd month no changes were presented – as proposed for the periods of 3 to 6 months. Is the proposal really to maintain the same exercises during the 3-month period?

Results

The authors presented results focused on the reproducibility of the tests and the effects of exercise on physical tests. As the central question of the introduction needs to be better justified, it is likely that the form or focus of the results should be improved.

Discussion and Conclusions

Lines 323 – 380

The discussion is speculative and goes beyond the results found in the study. What question do the authors want to answer? Is it possible for people to perform an exercise protocol?

The exercise protocol can cause changes in physical aspects. Could these improvements somehow be related to cognitive improvements?

What is new or what is the objective of the protocol of the PA component of the GINGER?

Reviewer 2 Report

Comments and Suggestions for Authors

I think this study is excellently framed– authors did a great job concisely operationalizing key variables (SCD), justifying a need, and articulating research aims. The materials and methods were also very clear– was satisfied with the defining of selected outcomes.

Sampling: Add a brief justification for exclusion criteria in section 2.1. Follow-up on screening process in results: how many did you exclude? This could be in a figure in supplemental materials. Are there other sample demographic variables to report beyond gender and age that you believe could be relevant?

In results, attrition of 50% of eligible participants post-screening is probably related to your RQ. As a reader, I want this to be attended to beyond “personal reasons unrelated to study” (line 303).

Struggling with the sample size of 3 for the measurements you’re reporting. With a sample size like this, I expect more depth of exploration around briefly-mentioned components of feasibility, such as teleconferencing familiarity (line 307), absence from sessions (line 317), participant satisfaction (line 319), and time constraints (line 412).

In a feasibility study, I also expect there to be some attention to context/environment, scalability, adaptability, even if in the limitations section. It is important to attend to these items as they pertain to your primary aim. 

It will strengthen your findings to contextualize this exploratory, small sample feasibility study with additional literature in the discussion section. How does this fit into existing intervention work in the area of physical activity to offset dementia, decelerate SCD?

Comments on the Quality of English Language

  • Tense changes in Section 2. Past tense?

  • Plural pronouns like “they” and “their” could replace “his/her” or “he/she” (lines 178-184)

  • Remove “etc.” (line 77)

  • “Enrolment” (line 113)

  • Extra indentation in lines 317, 325, 335.

  • In abstract, I can assume what “tailor-based” means but I’ve not seen that written before. Consider language like adaptable or tailorable?

Reviewer 3 Report

Comments and Suggestions for Authors

This study aimed  to present the developed protocol for the Physical Activity component of GINGER for people with SCD. This description should be improved and based on the literature review carried out. However, little is described from this literature review.

Please see other comments below…

ABSTRACT

P1L48. “...cognitive function..." - Cognitive function was assessed? If not, I believe that you can include this in conclusions...

INTRODUCTION

P2L67. “...while the thorough examination of their mental and cognitive function does not reveal cognitive deficits..” – Please explain this examination...

P2L68. “ Usually, these people are given the diagnosis of subjective cognitive decline (SCD).” – Reference... What are the ages and gender at which the appearance of SCD is predominant?

I believe the term "Physical activity" is used incorrectly throughout the text. The term "physical exercise" must be used...

MATERIALS AND METHODS

P3L115. “....utilizing Montreal Cognitive Assessment[PA2] (MoCA>26).” – Explain the test... Reference...

P3L123. “Α thorough literature review followed by a focus group study was undertaken.” – This review should be described in the article... Because it is from there that the exercise program was defined...

P3L140. “2.1.2. Outcome measures" - It is not understandable why no outcome evaluates cognitive status...

P4L185. “2.1.4. Intervention development" - Why was the program defined in this way?

P4L189. “One out of the two group sessions will be delivered via teleconference (in real time) to facilitate attendance." - And the other?

P4L195. “warm-up (5min), cool-down (5min)..." - Which exercises?

How was heart rate controlled?

Is it not clear which strength and stability exercises were chosen? Were they the same for all participants?

P5L200. “Repetitions of each RE will be done at an intensity of 50-80% based on..." - 50-80% of what?

RESULTS

P9L312. “3.5. Exercise Adherence and Satisfaction" -In the previous paragraphs you refer 13 participants, but here there are only 3? It's not clear...

CONCLUSIONS

Cognitive function was not assessed. The conclusions must be based on the data from the study...

Comments on the Quality of English Language

No comments.

Reviewer 4 Report

Comments and Suggestions for Authors

Thanks for the opportunity to review the paper "Physical Activity component of the Greek Interventional Geriatric Study to Prevent Cognitive Impairment and Disability (GINGER): protocol development and feasibility study". 

A very pertinent study in the field of older adults, considering the increased prevalence of health conditions such as dementia.

I make some suggestions for improvement and reformulation throughout the paper in sections.

Abstract

The objective is not clear in the abstract, they should clarify and rephrase the sentence to make it clearer. Because the conclusion in the abstract should respond to the objectives and you can't see what the objective is.

Introduction

A clear, objective and relevant introduction to the topic to be studied.

Materials and Methods

In this section I think it's important to clarify why you have two supervised groups and only one home-based group? We don't understand what the point is. Could you please rephrase and clarify?

Is the exercise protocol based on scientific studies? Who designed it? Based on what?

This information is important to include in the methods.

Results

They must standardise the number of decimal places in all tables.

Discussion

How important is online supervision? Can you clarify?

The combination of two supervised sessions and one session at home seemed ideal, why? What is the evidence for this?

The limitations of the study and future suggestions for the protocol remain to be mentioned. It would be pertinent to add this information.

Round 2

Reviewer 3 Report

Comments and Suggestions for Authors

The authors responded to all my comments satisfactorily and there was a clear improvement in the manuscript.

Reviewer 4 Report

Comments and Suggestions for Authors

Thank you for the opportunity to review this paper again.

The suggestions/changes requested have been accepted by the authors and have greatly improved the paper.